# Indirect sexual selection drives rapid sperm protein evolution in abalone

**Damien Beau Wilburn[1]\*, Lisa M Tuttle[2], Rachel E Klevit[2], Willie J Swanson[1]**

[1]Department of Genome Sciences, University of Washington, Seattle, United States; [2]Department of Biochemistry, University of Washington, Seattle, United States

**Abstract** Sexual selection can explain the rapid evolution of fertilization proteins, yet sperm proteins evolve rapidly even if not directly involved in fertilization. In the marine mollusk abalone, sperm secrete enormous quantities of two rapidly evolving proteins, lysin and sp18, that are stored at nearly molar concentrations. We demonstrate that this extraordinary packaging is achieved by associating into Fuzzy Interacting Transient Zwitterion (FITZ) complexes upon binding the intrinsically disordered FITZ Anionic Partner (FITZAP). FITZ complexes form at intracellular ionic strengths and, upon exocytosis into seawater, lysin and sp18 are dispersed to drive fertilization. NMR analyses revealed that lysin uses a common molecular interface to bind both FITZAP and its egg receptor VERL. As sexual selection alters the lysin-VERL interface, FITZAP coevolves rapidly to maintain lysin binding. FITZAP-lysin interactions exhibit a similar species-specificity as lysin-VERL interactions. Thus, tethered molecular arms races driven by sexual selection can generally explain rapid sperm protein evolution.

## Introduction

Genes associated with fertilization are often the fastest evolving in any genome (*Swanson and Vacquier, 2002*), and in mammals, spermatozoa-specific genes show the greatest divergence between species (*Torgerson et al., 2002*). While cooperation may be expected over conflict, differences in male and female reproductive strategies result in sexual arms races which can cause the rapid evolution of exaggerated sexual characters that have been hypothesized since Darwin as a driver of speciation (*Andersson, 1994*). Sexual selection acting on gamete recognition proteins is postulated to create reproductive barriers and facilitate speciation (*Arnold and Houck, 2016*; *Wilburn et al., 2017*), but sexual selection theory has not previously explained why sperm proteins that do not directly interact with the egg evolve rapidly. Here, we demonstrate that sexual selection can propagate through protein interaction networks and potentially drive global evolution of the sperm proteome.

The marine mollusk abalone is a classic system to study molecular barriers to hybridization (*Lewis et al., 1982*) and is the source of the first discovered pair of interacting reproductive proteins: sperm lysin and the egg vitelline envelope receptor of lysin (VERL) (*Swanson and Vacquier, 1997*). Animal eggs are surrounded by an extracellular barrier called the vitelline envelope (VE) that restricts the entry of sperm (the mammalian VE is referred to as the zona pellucida, ZP). VERL is a major component of the abalone VE which lysin dissolves by binding to repetitive domains within VERL (*Raj et al., 2017*; *Swanson and Vacquier, 1997*). Over millions of years, changes in VERL have resulted in positive sexual selection on lysin and a coevolutionary chase to maintain binding affinity. Extant lysins dissolve conspecific VEs more efficiently than those of closely related taxa, providing one mechanism of species-specific fertilization and a barrier to hybridization (*Swanson and Vacquier, 1997*; *Vacquier and Lee, 1993*). As VE dissolution is mediated by non-enzymatic lysin-VERL binding, the process is concentration dependent and sperm express enormous quantities of lysin (*Lewis et al., 1982*). A single male abalone can contain >1 gram of lysin, reflecting >0.1% of its total

**\*For correspondence:**
dwilburn@u.washington.edu

**Competing interests:** The authors declare that no competing interests exist.

body weight. Lysin is stored in a specialized secretory granule termed the acrosome. Based on electron microscopy (*Haino-Fukushima and Usui, 1986*; *Lewis et al., 1980*) we estimate that the acrosomal concentration of lysin is ~0.1–1.0 M, in stark contrast to saturation concentrations of ~0.001 M under in vitro conditions (*Wilburn et al., 2018*).

Lysin is not the only highly abundant, rapidly evolving protein in the abalone sperm acrosome. Another is sp18, a fusogenic paralog of lysin that likely mediates plasma membrane fusion between egg and sperm (*Swanson and Vacquier, 1995*). The receptor of sp18 is unknown, but given its interaction with the abalone egg, its accelerated evolution is likely due to sexual selection (*Aagaard et al., 2010*). Sp18 is nearly as abundant as lysin, so it must also be packaged at high concentrations, yet its fusogenic properties make it even less soluble than lysin (*Kresge et al., 2001*). Recently, a new family of small acrosomal proteins termed sperm protein 6 kDa (sp6) was discovered by shotgun transcriptomics and proteomics (*Palmer et al., 2013*). While also rapidly evolving and hypothesized to evolve via sexual selection, initial efforts to identify an egg binding partner for sp6 were unsuccessful (*Palmer, 2013*). However, while lysin and sp18 are highly positively charged proteins (+12 to +24), isoforms of sp6 are highly anionic (−6 to −16) and include an N-terminal poly-aspartate region of variable length (1–11 residues). Given this charge complementarity, we hypothesized that sp6 may facilitate packaging of lysin and sp18 inside the sperm acrosome. We demonstrate that the rapid evolution of sp6 is due to intra-sperm protein coevolution with lysin and sp18 to allow for their dense storage in the acrosome via novel Fuzzy Interacting Transient Zwitterion (FITZ) complexes. Heterodimers of lysin-sp6 or sp18-sp6 form through hydrophobic interactions, and these heterodimers polymerize into large particles (diameter >100 nm) through ionic interactions of the complementary positive and negative charges. Upon secretion of the acrosomal contents into highly ionic seawater, FITZ complexes are disrupted, and the dispersal of lysin and sp18 facilitating fertilization. In light of its newly identified function, we have named sp6 the FITZ Anionic Partner (FITZAP).

## Results

Different species of abalone express different numbers of FITZAP isoforms named for the length of the N-terminal poly-aspartate region (*Palmer, 2013*). In red abalone (*Haliotis rufescens*), two isoforms (FITZAP-4D and FITZAP-8D) result from alternative splicing of different versions of exon 1 (signal peptide and the N-terminus with the poly-aspartate region) with a common exon 2 (C-terminus) (*Figure 1—figure supplement 1*). Each isoform was purified using strong anion exchange (SAX) chromatography and reverse-phase high-performance liquid chromatography (RP-HPLC), with mass spectrometry revealing that both isoforms were smaller than their cDNA open reading frame predicted (~3–4 kDa vs ~6 kDa). The observed masses are consistent with proteolytic processing of FITZAP by the Golgi enzymes furin and carboxypeptidase B (*Figure 1—figure supplement 2*). Despite their high net positive charges, both lysin and sp18 co-eluted with FITZAP at high-salt concentrations by SAX chromatography (*Figure 1*). Particularly striking is that sp18 (+22) eluted at higher salt concentrations than lysin (+12), which mirrors the elution profiles of FITZAP-4D (−10) and FITZAP-8D (−8), respectively, suggesting isoform-specific interactions. While purified lysin showed no affinity for anion exchange resin, its elution was retarded when mixed with either FITZAP-4D or FITZAP-8D in vitro, albeit less dramatically than the ex vivo samples (*Figure 1—figure supplement 3*). This is likely a consequence of sample preparation and the extraordinary concentration of lysin and FITZAP inside the acrosome compared to in vitro reconstitutions, allowing more complexes to persist during the chromatography. Together, these findings support that in vivo interactions likely enable co-purification of lysin/sp18 and FITZAP from sperm lysate.

Nuclear magnetic resonance (NMR) spectroscopy was used to investigate how FITZAP interacts with the cationic fertilization proteins. We focused on lysin and FITZAP-8D because (A) they showed the strongest coelution by SAX chromatography (*Figure 1*), (B) lysin is more soluble than sp18 (*Aagaard et al., 2010*; *Swanson and Vacquier, 1995*), (C) interactions with its egg receptor have been characterized (*Raj et al., 2017*; *Wilburn et al., 2018*), and (D) a solution structure has been determined (*Wilburn et al., 2018*). Chemical shift analysis of FITZAP-8D revealed that it is an intrinsically disordered protein (IDP); even when bound to lysin, FITZAP-8D remained highly dynamic and adopted no regular secondary structure (*Figure 2—figure supplements 1* and *2*). Thus, lysin and FITZAP form a fuzzy complex: protein complexes that exist in an ensemble of different interchanging configurations (*Figure 2*). Formation of the fuzzy complex is primarily due to packing between

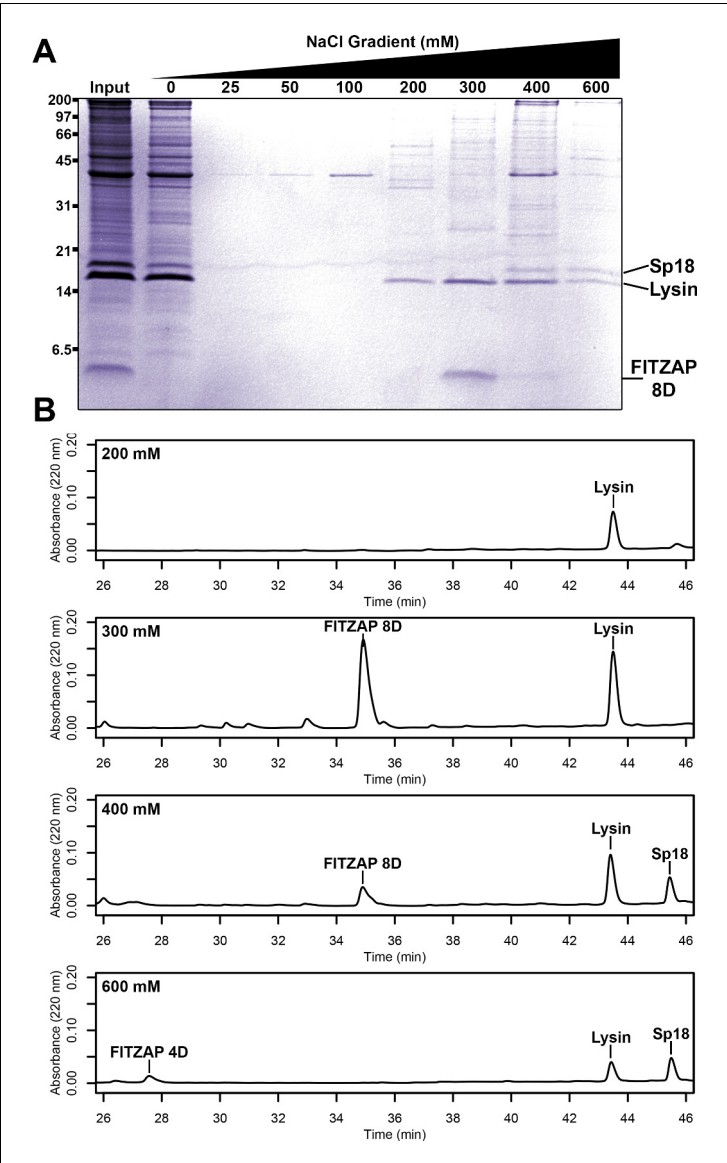

**Figure 1.** Co-purification of FITZAP and cationic acrosomal proteins by anion exchange. (**A**) SDS-PAGE of proteins from red abalone sperm lysate separated by anion exchange. Despite their high net positive charges, lysin and sp18 co-eluted with FITZAP isoforms in the highest salt fractions. (**B**) RP-HPLC analysis of select anion exchange fractions which contained lysin. The peak of lysin elution at 300 mM coincides with the peak elution of FITZAP-8D, with both sp18 and FITZAP-4D eluting under higher salt conditions.

The online version of this article includes the following source data and figure supplement(s) for figure 1:

**Figure supplement 1.** Genomic structure of FITZAP locus in red and disk abalone.

**Figure supplement 1—source data 1.** Sequence alignment of FITZAP loci from red and disk abalone draft genomes.

**Figure supplement 2.** Purification and mass spectral analysis of natural red FITZAP-4D and 8D.

**Figure supplement 3.** SAX chromatography of lysin and FITZAP in vitro.

hydrophobic amino acids in residues 21–29 of FITZAP (*Figure 3A*) and an exposed hydrophobic face on lysin near the nexus of the N- and C-termini. Covering this hydrophobic patch with FITZAP imparts a high net negative charge to this region of lysin, likely explaining the affinity of lysin to anion exchange resins when FITZAP is present. Significantly, the FITZAP-binding region of lysin is also the same surface that recognizes its egg receptor VERL, based on both chemical shift perturbations (*Figure 2A*) and paramagnetic relaxation enhancement (PRE) data (*Figure 2—figure*

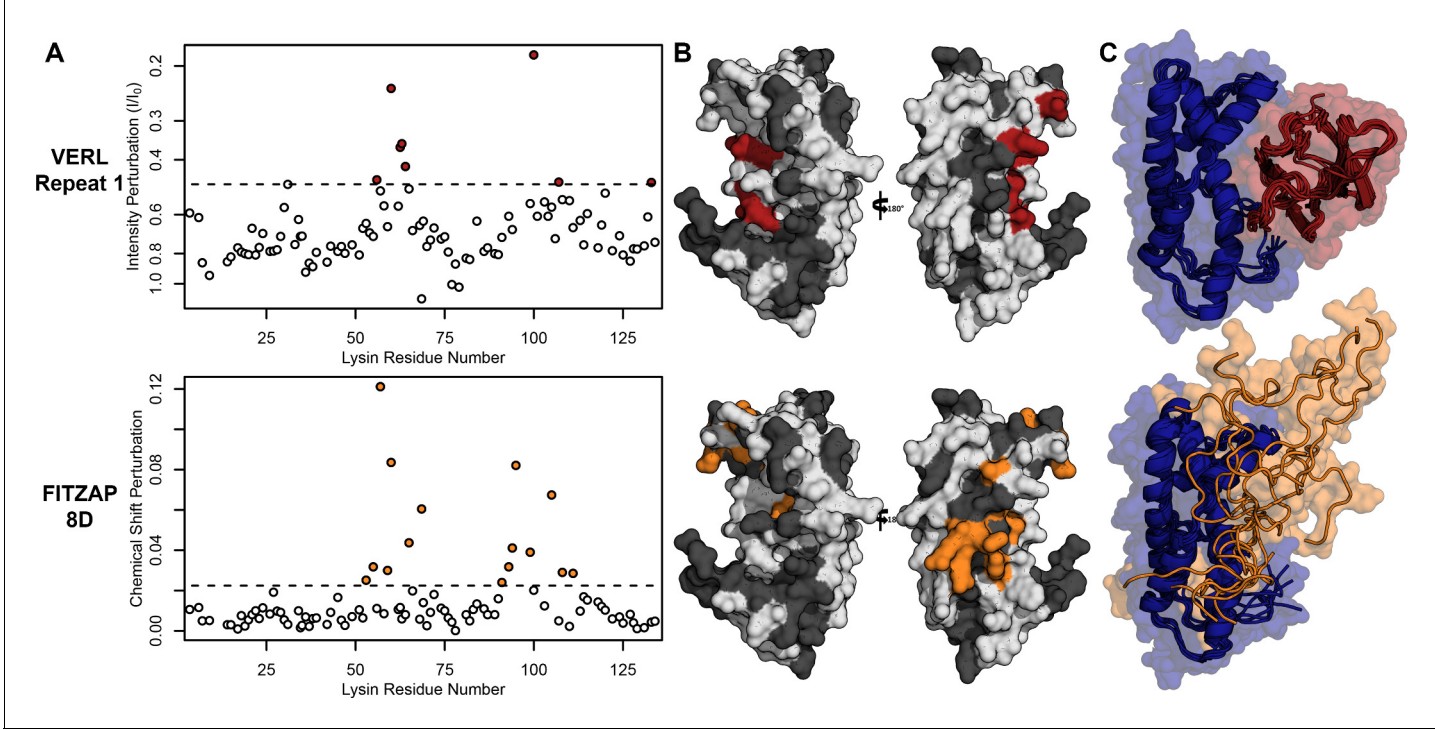

**Figure 2.** Lysin recognizes VERL and FITZAP through a common binding interface. (**A**) NMR perturbation of red lysin upon conspecific binding of either VERL repeat 1 (data from *Wilburn et al., 2018*) or FITZAP-8D (1.5 molar equivalents). Two regions near residue 60 and residue 100 of lysin are perturbed in both cases (dashed line: median + 1.5 * interquartile range). (**B**) Mapping of perturbation onto a lysin solution structure (PDB 5utg) shows spatial clustering of these two regions to a single binding surface of lysin. (**C**) Docking of VERL repeat 1 (based on PDB 5mr3 by *Raj et al., 2017*) and FITZAP-8D (based on restraints from paramagnetic relaxation enhancement) supports that FITZAP is an intrinsically disordered protein that shields the VERL binding interface of lysin.

The online version of this article includes the following source data and figure supplement(s) for figure 2:

**Source data 1.** Lysin NMR perturbation values for VERLr1 and FITZAP-8D.

**Figure supplement 1.** FITZAP is intrinsically disordered when free and bound to lysin.

**Figure supplement 1—source data 1.** FITZAP chemical shift indices.

**Figure supplement 2.** Lysin binding causes few changes in FITZAP $^{13}$C chemical shifts.

**Figure supplement 3.** Intermolecular PRE measurements between $^{15}$N-lysin and FITZAP-8D.

**Figure supplement 3—source data 1.** PRE measurements of lysin-FITZAP interactions.

*supplement 3*). This exposed hydrophobic surface is likely why purified lysin has a much lower in vitro solubility limit compared to the acrosome where FITZAP is also highly abundant.

Given a single interaction surface for two different binding partners, lysin must separately recognize FITZAP inside the acrosome and, upon secretion, VERL at the VE. As abalone is a marine mollusk, we postulated that local inorganic salt concentrations may play an important role. The total salt concentration of seawater is ~500 mM, yet the intracellular environment of marine animals is less concentrated. Osmolality is maintained in marine vertebrates by active transport of water and ions, whereas many invertebrates such as abalone are osmoconformers and use high intracellular concentrations of free amino acids, betaines, and other highly soluble metabolites to achieve osmolality (but not isotonicity) (*Venter et al., 2018*). While the exact intracellular concentration of inorganic ions in abalone sperm is unknown, it is likely much lower than seawater based on data from sea urchin gametes (*Horthschild and Barnes, 1953*; *Rodríguez and Darszon, 2003*) and other abalone tissues (*Jia and Liu, 2018*). The inorganic salt concentration may be even lower in the acrosome where the acrosomal proteins – given their extraordinary concentrations and net charges – may themselves serve as osmolytes. To compare how lysin-FITZAP interactions are influenced by differing environmental contexts, we performed biophysical experiments under low (150 mM NaCl) and high (500 mM NaCl) salt concentrations that approximate the intracellular or seawater environments,

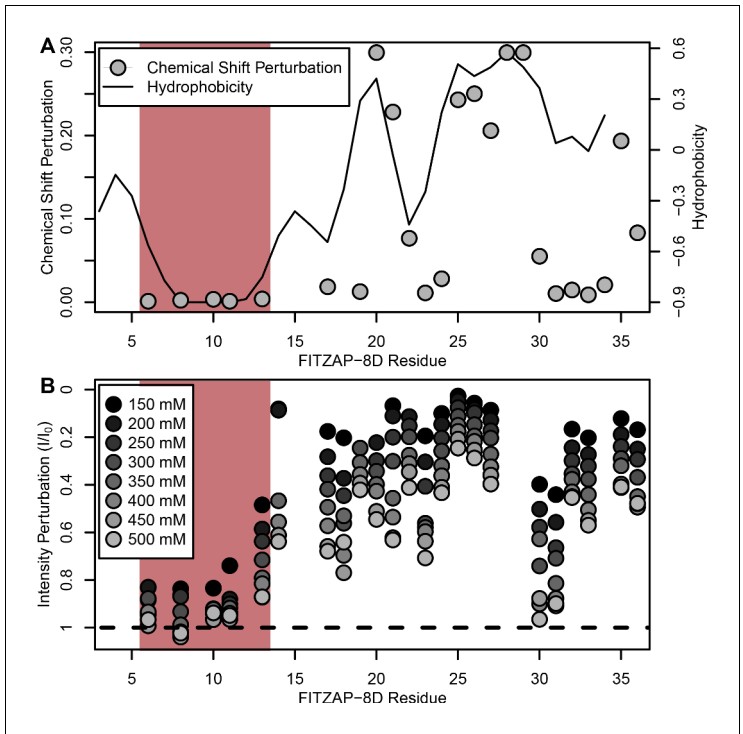

**Figure 3.** NMR perturbation of FITZAP-8D upon lysin binding. (**A**) Backbone amide chemical shift perturbation of FITZAP-8D upon addition of lysin under seawater conditions (500 mM NaCl). The solid line denotes the relative hydrophobicity of the sequence which correlates well with the chemical shift perturbation, supporting that FITZAP-lysin interactions under high-salt conditions are driven by hydrophobic packing. The poly-aspartate region is highlighted in red and shows no chemical shift perturbation under seawater conditions. (**B**) Intensity perturbation of FITZAP-8D by binding of lysin under different ionic strengths ranging from approximately intracellular levels (150 mM) to seawater conditions (500 mM). Differences in intensity perturbation as a function of salt are in part driven by the salt dependence on lysin-FITZAP $K_d$ (*Figure 5*), but the weak perturbation observed in the poly-aspartate region only under low salt conditions support that these residues are involved in some form of molecular interaction with lysin (likely salt bridges as part of FITZ complexes) only under intracellular conditions.

The online version of this article includes the following source data and figure supplement(s) for figure 3:

**Source data 1.** FITZAP chemical shift perturbations by lysin binding.
**Source data 2.** FITZAP intensity perturbation by lysin at differing salt concentrations.
**Figure supplement 1.** FITZ complexes experience different exchange rates in intracellular vs seawater conditions.
**Figure supplement 1—source data 1.** Lysin intensity and chemical shift perturbation by FITZAP binding at differing salt concentrations.

respectively. Under both conditions, the hydrophobic patch of FITZAP interacts with lysin, yet NMR intensity perturbation of the poly-aspartate region was only observed under low-salt concentrations where intermolecular salt bridges may form (*Figure 3B*). Differences in NMR perturbation establish that lysin and FITZAP undergo slower subunit exchange under low-salt concentrations (*Figure 3—figure supplement 1*). Equimolar mixtures of lysin and FITZAP-8D under low-salt conditions form extremely large oligomers with an average diameter of ~400 nm (compared to a mean lysin diameter of ~6 nm); these large particles are not present at high-salt concentrations (*Figure 4*). Our data establish that lysin and FITZAP associate hydrophobically to form fast-exchanging, fuzzy hetero-dimers that are essentially zwitterionic. These heterodimers that we call Fuzzy Interacting Transient Zwitterions (FITZs) can form intermolecular salt bridges that allow tight packaging under intracellular-like conditions. Upon secretion into seawater, the FITZ complexes are disrupted and subunit exchange rate increases, allowing lysin to be rapidly liberated from FITZAP, permitting interactions with VERL via a common binding interface. The formation and dissolution of FITZ complexes based on the environmental context provides a mechanism for exceptionally high packaging concentrations

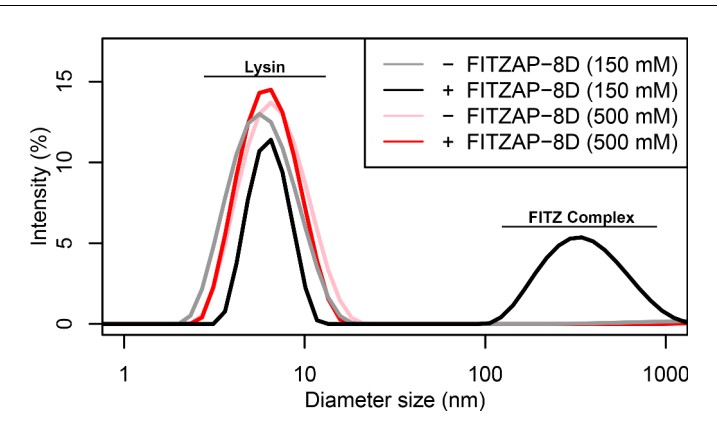

**Figure 4.** FITZ complex formation is dependent on both FITZAP and the ionic environment. Dynamic light scattering measurements of lysin with and without equimolar FITZAP-8D at intracellular (150 mM) and seawater (500 mM) salt conditions. Lysin and FITZAP associate into FITZ complexes with mean diameter of ~400 nm only under intracellular salt conditions.

of lysin inside the sperm acrosome as well as its rapid dispersal in seawater when fertilization may be imminent.

Lysin and VERL are rapidly coevolving sperm and egg proteins that exhibit species-specific interactions (*Wilburn and Swanson, 2016*). Given the common binding interface, we postulated that there may be similar coevolution between lysin and FITZAP. While such coevolution was not detected using sequence-based approaches (*Figure 5—figure supplement 1*), the small size and intrinsic disorder of FITZAP can reduce statistical power of such analyses. A functional consequence of lysin and FITZAP coevolution would be species-specific interactions where the proteins from the same species have higher binding affinities compared to heterospecific pairs, as observed between lysin and VERL. Using fluorescence polarization, binding affinities were measured between lysin and all FITZAP isoforms for three abalone species (red, disk, and green). Like red abalone, green abalone has two FITZAP isoforms but with shorter poly-aspartate regions (1D and 4D), while disk abalone has a single FITZAP isoform with an even longer poly-aspartate array (11D). Lysin had greater affinity for the high-D FITZAP isoforms compared to low-D forms. For all three species, lysin bound the conspecific high-D isoforms of FITZAP with equilibrium dissociation constants of ~1–2 µM under intracellular salt conditions. Except for disk lysin and red FITZAP-8D, all cases of heterospecific binding were significantly weaker, supporting lysin-FITZAP coevolution (*Figure 5A*). Both hydrophobic and ionic interactions likely contribute to these higher affinity complexes, yet the specificity of disk FITZAP-11D (with the longest poly-aspartate array) to disk lysin supports that electrostatic attraction alone is not sufficient to explain FITZ complex formation. Under seawater conditions, there was a $\geq$ 20 fold decrease in conspecific binding affinity (*Figure 5B*), consistent with the liberation of lysin from FITZ complexes after its release from the acrosome.

As sp18 likely facilitates the fusion of egg and sperm plasma membranes (*Swanson and Vacquier, 1995*), it is more hydrophobic than lysin and may therefore be even more reliant on a partner for storage and dispersal. Lysin and sp18 are paralogs with similar tertiary structures that have subfunctionalized following gene duplication (*Kresge et al., 2001*; *Swanson and Vacquier, 1995*), so the multiple FITZAP isoforms may also have subfunctionalized to bind these different paralogs. As lysin showed greater affinity for high-D FITZAP isoforms (*Figure 5A*), we hypothesized that sp18 may be coevolving with low-D isoforms. We indeed observe correlated rates of molecular evolution between sp18 and low-D FITZAP isoforms, consistent with coevolution (*Figure 5—figure supplement 1*). While the highly fusogenic sp18 is mostly insoluble when purified (*Aagaard et al., 2010*; *Swanson and Vacquier, 1995*), we were able to measure conspecific affinities between sp18 and FITZAP isoforms from green abalone. While green sp18 also bound green FITZAP-4D more tightly than FITZAP-1D (*Figure 5C*), its relative affinity for 4D over 1D (~3.5X) is substantially less compared to the relative affinity of 4D to 1D for lysin (>100X). For our anion exchange experiments in red

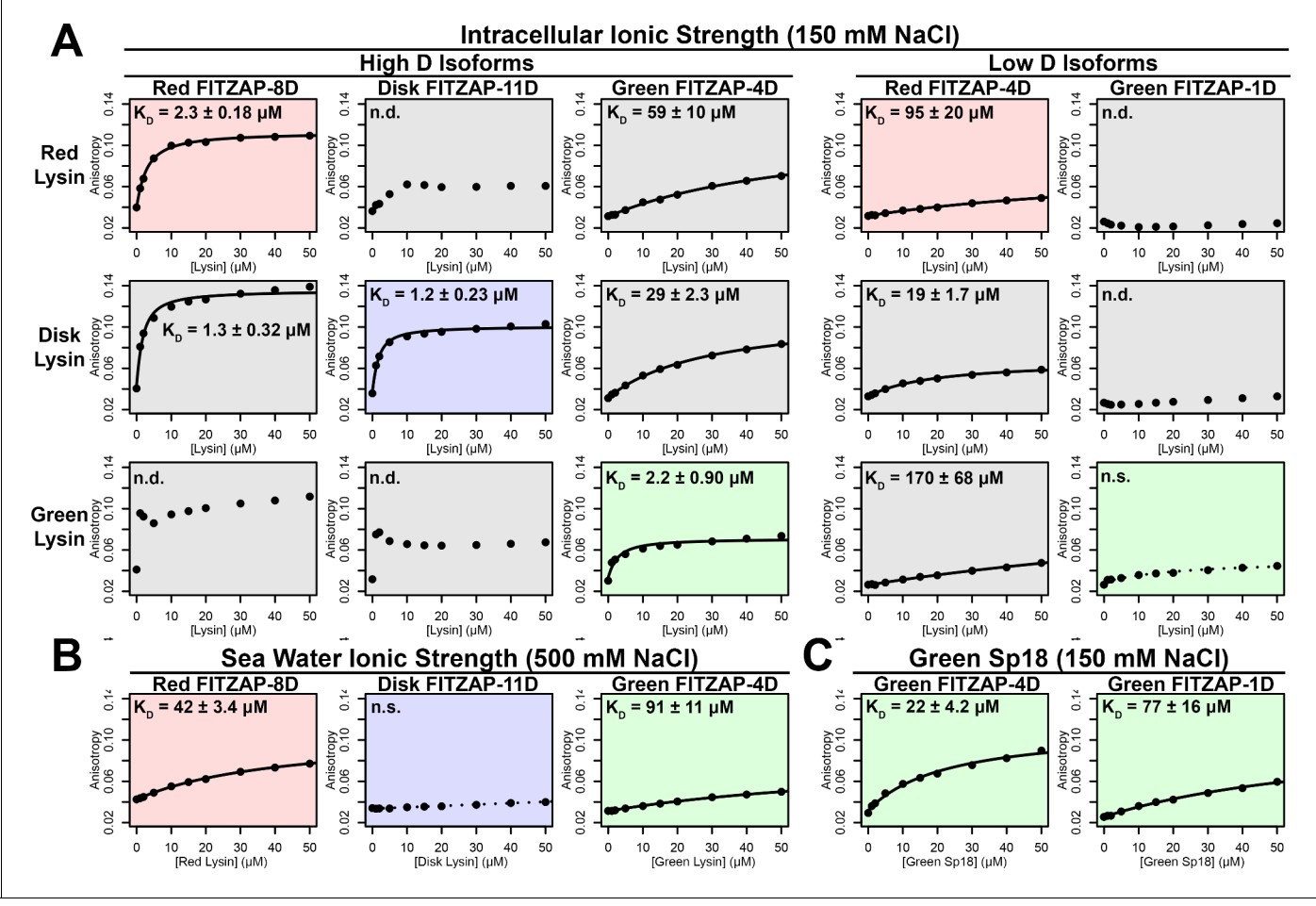

**Figure 5.** FITZAP interactions with cationic acrosomal proteins within and between species. (**A**) Lysin-FITZAP interactions were measured by fluorescence polarization under approximately intracellular salt conditions for all combinations within and between species of red, disk, and green abalone (conspecific interactions are shaded as red, blue, and green, respectively). Low micromolar binding affinities were observed for conspecific interactions of lysin with high-D FITZAP isoforms. (**B**) Binding affinities between conspecific lysin and FITZAP high-D isoforms are substantially reduced under extracellular seawater conditions. (**C**) Green sp18 shows tighter binding to conspecific FITZAP-4D compared to FITZAP-1D; however, the relative affinity of sp18 for 4D over 1D (77/22 = 3.5 X) is greater than that of lysin (>100 X), suggesting that sp18 has higher preference for low D isoforms compared to lysin. (n.s. = not significant at p<0.05; n.d. = not determined if anisotropy was not monotonically positive and consistent with single-state binding; $K_d$ reported as mean ± standard error).

The online version of this article includes the following figure supplement(s) for figure 5:

**Figure supplement 1.** Test of sequence coevolution between abalone reproductive genes by correlated branch $d_N/d_S$.

abalone, we observed tighter co-elution between lysin/FITZAP-8D and sp18/FITZAP-4D (**Figure 1**). Therefore, both evolutionary and biochemical evidence support that FITZAP isoforms have subfunctionalized to respond to the divergent evolutionary trajectories of lysin and sp18.

## Discussion

We propose a system of tethered coevolution between VERL, lysin, and FITZAP where VERL imposes direct sexual selection on lysin and indirect sexual selection on FITZAP (summarized in **Figure 6**). As with any coevolving system, there is likely reciprocity with all binding partners imposing some form of selection on one another; however, we choose to focus on the unidirectional case of VERL influencing lysin influencing FITZAP for several reasons. First, sexual selection theory has emphasized 'female choice,' because the higher energetic cost of oocytes in most species favor greater mate selectivity (**Arnold and Houck, 2016**). These assumptions are most apt when discussing genes

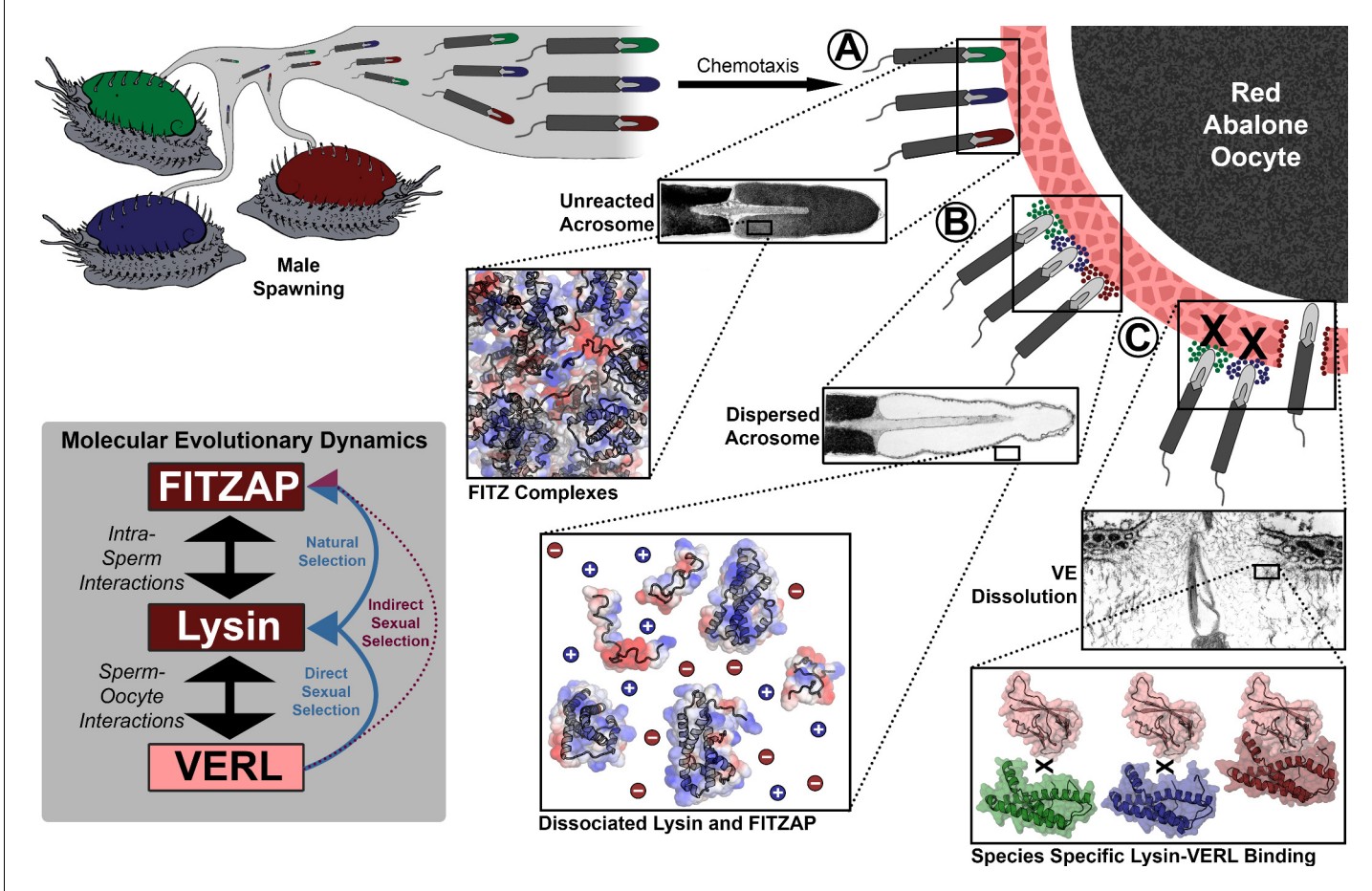

**Figure 6.** Rapid evolution of FITZAP is due to indirect sexual selection from VERL. Male abalone have overlapping habitat ranges and spawning periods such that there is opportunity for hybridization to occur. Sperm are attracted to eggs via chemotaxis and bind to the vitelline envelope (A). At this stage, the acrosome is still intact (*Lewis et al., 1982*) with lysin and FITZAP tightly packaged as FITZ complexes. Binding to the VE causes the sperm to acrosome react (B), releasing its contents into highly ionic seawater which disperses lysin and FITZAP. Liberated lysin can now dissolve the VE by binding VERL domains (C); these interactions are species-specific and provide one barrier to hybridization (*Swanson and Vacquier, 1997*). Because the rapid evolution of lysin is driven by direct sexual selection to maintain sperm-egg interactions, and FITZAP is co-evolving to bind the same interface, VERL imposes indirect sexual selection onto FITZAP. Microscopy images adapted from *Lewis et al. (1982)* and *Sakai et al., 1982*.

involved in fertilization. Second, lysin evolves ~5X faster than its coevolving regions of VERL (*Galindo et al., 2003*), suggesting that it is experiencing greater directional selection than VERL. Lysin, but not VERL, is also monomorphic within some abalone populations and experienced recent selective sweeps (*Clark et al., 2009*). Third, in sea urchins (another organism with broadcast spawning), longitudinal measurements of allele frequencies for gamete recognition proteins support that female proteins will shift to lower affinity interactions in response to increased polyspermy risk, and high-affinity binding is restored by adaptation of male proteins (*Levitan et al., 2019*). This suggests that evolutionary dynamics of lysin are more responsive to VERL than vice versa. Fourth, as FITZAP is an IDP whose function is to facilitate the storage and dispersal of fertilization proteins, we anticipate reduced conservation on its primary sequence from intramolecular epistasis, with its evolution mostly being driven by directional selection imposed by its binding partners.

Because IDPs lack a single favored conformation, they experience less purifying selection (or slower evolution relative to genetic drift) to maintain a tertiary fold and have lower sequence conservation (*Brown et al., 2011*). Beyond relaxed purifying selection and greater rates of genetic drift, IDPs also experience more positive selection compared to structural domains (*Afanasyeva et al., 2018*), presumably in response to coevolution with their binding partners. We expect FITZAP to

respond more to selection from lysin than vice versa. It has been suggested that intrinsic disorder may be a mechanism of adaptation to shifts in environmental conditions; for example, host-changing parasites have higher genome-wide levels of predicted protein disorder compared to obligate intracellular parasites and endosymbiotes (*Pancsa and Tompa, 2012*). The change in ionic strength experienced by secreted proteins of most marine animals is likely an extreme example of such shifts between chemical environments. The intrinsic disorder of FITZAP may be crucial to its apparent structural versatility and high evolvability in response to a rapidly coevolving partner.

Across animals, plants, and microbes, genes associated with fertilization usually evolve faster than the rest of the genome (*Swanson and Vacquier, 2002*; *Wilburn et al., 2017*; *Wilburn and Swanson, 2016*). Like macroscopic secondary sex characters, direct sexual selection can drive elaboration of molecular phenotypes such as the extraordinary abundance of lysin and sp18. Combined with sequence differences that yield species-specific interactions with coevolving receptors, these proteins are one of many reproductive barriers that likely contribute to speciation. Given the necessity of fertilization for sexually reproducing taxa, few selective pressures are likely stronger than sexual selection, and even its indirect effects through coevolutionary networks are likely substantial. In this report, we demonstrated how indirect sexual selection drives rapid evolution of a sperm protein not associated with fertilization. Indirect selection may further propagate throughout the sperm proteome and be a general mechanism to explain accelerated gametic protein evolution.

# Materials and methods

## Key resources table

| Reagent type (species) or resource | Designation | Source or reference | Identifiers | Additional information |
|---|---|---|---|---|
| Strain, strain background (*Escherichia coli*) | 5-alpha | New England Biolabs | C2987 | Chemically competent cells for cloning and sequencing |
| Strain, strain background (*Escherichia coli*) | Rosetta2 | Novagen | 71400 | Chemically competent cells for protein expression |
| Biological sample (*Haliotis rufescens*) | red abalone sperm | The Abalone Farm | | Freshly isolated from male abalone testis |
| Biological sample (*Haliotis fulgens*) | green abalone lysin | Collected from 32°51'00' N, 117°16'34' W | | Purified from sperm by cation exchange chromatography, and provided by Vic Vacquier |
| Recombinant DNA reagent | pCR4-TOPO | Thermo-Fisher | K457540 | Cloning of DNA for sequencing |
| Recombinant DNA reagent | pET11d | Novagen | 69439 | Cloning of DNA for protein expression |
| Recombinant DNA reagent | MBP-FITZAP | This paper | | See *Supplementary file 1* |
| Sequence-based reagent | FITZAP F' | This paper | | 5'-ATGAGGGGTTRTTCTAATT-3' |
| Sequence-based reagent | Sp18 F' | This paper | | 5'-GGAAACAGTATGAGGTYTTTGSTGCTT-3' |
| Peptide, recombinant protein | MBP-TEV protease | Sigma-Aldrich | T4455 | |
| Commercial assay or kit | BCA Protein Assay | Pierce | 23225 | |
| Software, algorithm | RStudio v.1.0.136 | https://rstudio.com/ | RRID: SCR_000432 | |

*Continued on next page*

*Continued*

| Reagent type (species) or resource | Designation | Source or reference | Identifiers | Additional information |
|---|---|---|---|---|
| Software, algorithm | RAxML v8.2.12 | https://cme.h-its.org/exelixis/software.html | RRID: SCR_006086 | |
| Software, algorithm | Fast Statistical Alignment (FSA) v.1.15.9 | http://fsa.sourceforge.net/ | RRID: SCR_016114 | |
| Software, algorithm | PAML v4 | http://abacus.gene.ucl.ac.uk/software/paml.html | RRID: SCR_014932 | |
| Software, algorithm | NMRFAM-Sparky | https://nmrfam.wisc.edu/nmrfam-sparky-distribution/ | | |
| Software, algorithm | Xplor-NIH v.2.48 | https://nmr.cit.nih.gov/xplor-nih/ | | |
| Software, algorithm | TALOS-N | https://spin.niddk.nih.gov/bax/software/TALOS-N/ | | |
| Software, algorithm | PyMOL v.1.8 | https://github.com/schrodinger/pymol-open-source | RRID: SCR_000305 | |

## Purification and mass spectral characterization of natural FITZAP

Natural FITZAP was purified and characterized based on methods modified from *Palmer et al. (2013)*. Briefly, sperm were collected by dissection of testes from red abalone (*Haliotis rufescens*) and lysed by trituration in 1% Triton X-100 (w/v)/250 mM NaCl/2 mM EDTA/10 mM MES, pH 6, and centrifuged at 3200 x *g*, 8°C for 30 min. The supernatant was applied to a 10 mL CM52 cellulose column (Whatman, Maidstone, UK) to remove lysin and other positively charged proteins. The FITZAP-enriched flow through was diluted with 2 volumes of 50 mM Tris, pH 8, applied to a 10 mL Q sepharose column (Sigma-Aldrich, St. Louis, MO), and protein fractions collected by gravity flow with a stepwise NaCl gradient buffered with 20 mM Tris, pH 8. Fractions were analyzed by 15% Tris-Tricine SDS-PAGE (*Schägger and von Jagow, 1987*), and FITZAP localized to fractions with ≥300 mM NaCl. These fractions were pooled and concentrated using 3 kDa centrifugal ultrafilters (EMD-Millipore, Billerica, MA), and individual components purified by reverse phase high performance liquid chromatography (RP-HPLC) using a Vydac C18 column (0.46 × 15 cm; Hichrom, Berkshire, UK) that was eluted from 0 to 70% acetonitrile in 0.1% trifluoracetic acid at 1% acetonitrile per minute. Individually purified proteins were analyzed by LC/MS-MS using data-dependent acquisition on an LTQ Velos tandem mass spectrometer (Thermo Scientific, Waltham, MA) for determination of intact protein and fragment ion masses.

## Sequence analysis of FITZAP and estimation of molecular evolutionary rates

Draft genome assemblies are available for disk abalone (*Haliotis discus*) (*Nam et al., 2017*) and red abalone (*H. rufescens*) (*Masonbrink et al., 2019*). The contigs or scaffolds containing FITZAP exons were identified by performing BLAST searches (*Camacho et al., 2009*) using FITZAP open reading frames as queries. These genomic regions were extracted with an additional 5 kb of flanking sequence on both the 5' and 3' ends, and aligned using fsa v1.15.9 (*Bradley et al., 2009*). For molecular evolutionary analysis, available cDNA sequences for lysin, VERL, sp18, and FITZAP were downloaded from Genbank (Accession # M34388, M59969-M59972, M98875, AF453553, AF490761-AF490763, AF490765, AF490766, L36552, L36554, L36589, KC752594, KC752595, KC752597-KC752602). We additionally sequenced sp18 cDNA by RT-PCR from black abalone (*Haliotis cracherodii*), flat abalone (*Haliotis wallallensis*), and disk abalone (*H. discus*), and FITZAP from flat abalone. Abalone testis cDNA was provided by Jan Aagaard, and RT-PCR was performed using an oligo-dT

reverse primer using a sp18 specific (5'-GGAAACAGTATGAGGTYTTTGSTGCTT-3') or FITZAP-specific (5'-ATGAGGGTTRTTCTAATT-3') forward primer. PCR products were cloned into a pCR4-TOPO vector (Invitrogen), transformed into 5-alpha competent *E. coli* (New England Biolabs, Ipswich, MA), and plasmid DNA from at least four clones of each transformation were supplied to Eurofins Scientific (Louisville, KY) for Sanger sequencing. No sequence variation was observed between clones, and these sequences have been deposited into Genbank (Accession # MN102340-MN102343). A maximum likelihood gene tree was constructed with RAxML v8.2.12 (*Stamatakis, 2014*) using the PROTGAMMALG substitution model and a concatenation of protein sequences from lysin, VERL, and sp18 from six abalone species (red, flat, disk, black, pinto (*Haliotis kamtschatkana*), and green (*Haliotis fulgens*)) aligned by Clustal Omega (*Sievers et al., 2011*), and used as a representative of the likely species tree. Based on methods by *Clark et al. (2009)*, support for co-evolution between protein coding genes was evaluated by weighted linear regression using branch $d_N/d_S$ values estimated with PAML v4 (*Yang, 2007*) for each of the four genes (lysin, VERL, sp18, and FITZAP), with FITZAP further divided into low-D and high-D isoforms.

## Cloning and expression of recombinant FITZAP

Recombinant FITZAP was expressed in *E. coli* and purified to near homogeneity using multiple chromatography steps. Given the low molecular weight of different FITZAP isoforms, expression in *E. coli* required fusion to a larger carrier protein from which FITZAP could be removed by enzymatic proteolysis and purified. FITZAP isoforms from red, disk, and green abalone were genetically fused by PCR with a maltose binding protein (MBP) cassette containing an N-terminal 6xHis tag and a C-terminal linker sequence followed by a tobacco etch virus (TEV) protease cleavage site (see *Supplementary file 1* for sequences). Recombinant FITZAP proteins included an Amino Terminal Cu- and Ni-binding tag (ATCUN) to facilitate protein purification, improve TEV proteolysis (*Kapust et al., 2002*), and permit collection of NMR PRE constraints (*Donaldson et al., 2001*). The combined MBP-FITZAP construct was cloned into the pET11d expression vector (Novagen, San Diego, CA), transformed into Rosetta2 chemically competent *E. coli* (EMD-Millipore, Billerica, MA) which express additional tRNA genes for Lys and Arg that are abundant in abalone genes, and clones validated by Sanger sequencing (Eurofin Genomics, Louisville, KY). For expression of unlabeled FITZAP, *E. coli* clones were cultured in LB media supplemented with 100 µg/mL ampicillin and 34 µg/mL chloramphenicol at 37°C, 250 rpm; when cultures reached an optical density at 600 nm (OD600) of ~0.6, recombinant protein expression was induced by addition of IPTG to a final concentration of ~100 µM, cells harvested by centrifugation after 3.5 hr, and stored at −20°C. For expression of isotopically labeled FITZAP, methods were adapted from *Wilburn et al. (2018)*. Briefly, *E. coli* clones expressing MBP-FITZAP were cultured in LB media supplemented with 100 µg/mL ampicillin and 34 µg/mL chloramphenicol at 37°C, 250 rpm until the OD600 reached ~0.4; cells were then collected by centrifugation, concentrated 4-fold into M9 media with 20 µM FeSO$_4$ and 100 µg/mL ampicillin without carbon or nitrogen sources, and maintained at 37°C, 250 rpm for 35 min to deplete the cells of free unlabeled amino acids; cultures were then supplemented with 3 g/L ammonium sulfate ($^{14}$N or 98% $^{15}$N) and 4 g/L glucose ($^{12}$C or 99% $^{13}$C) for 35 min to regenerate amino acid stores with appropriate isotopes; then expression was induced by addition of IPTG to a final concentration of ~100 µM, cells harvested by centrifugation after 3.5 hr, and stored at −20°C. Cell pellets were then lysed by sonication in 1% octylthioglucoside/50 mM NaCl/50 mM Tris, pH 8, then supplemented with 0.2 mg/mL lysozyme for 30 min, centrifuged @ 3.2 k x *g* for 2 hr, the supernatant clarified by passage through a 0.2 µm PES filter, and the filtrate applied to a 10 mL Ni-NTA column (Pierce, Rockford, IL) equilibrated in 500 mM NaCl/20 mM Tris/1 mM imidazole, pH 8. The column was subsequently washed with increasing concentrations of imidazole in 500 mM NaCl/20 mM Tris, pH 8: six column volumes (CVs) at 1 mM imidazole, 2 CVs at 20 mM, 1 CV at 40 mM, 1 CV at 60 mM, and MBP-FITZAP eluted using 3 CVs at 200 mM imidazole. The elution fraction was buffer exchanged using a YM30 centrifugal ultrafilter (Millipore, Billerica, MA) into 100 mM NaCl/20 mM Tris, pH 8, supplemented with TEV Protease (Sigma-Aldrich) at an enzyme:substrate ratio of ~1:500 by mass, and incubated overnight at room temperature with gentle mixing. Following proteolysis, precipitate was removed by centrifugation at 3.2 k x *g* for 30 min, and the supernatant applied to a 10 mL Ni-NTA column equilibrated in 500 mM NaCl/20 mM Tris/1 mM imidazole, pH 8. The column was washed with 3 CVs of 500 mM NaCl/20 mM Tris/1 mM imidazole, pH 8, a FITZAP-enriched fraction eluted using 3 CVs of 500 mM NaCl/20 mM Tris/20 mM imidazole, pH 8, and MBP/MBP-FITZAP

enriched fraction eluted using 500 mM NaCl/200 mM Tris/1 mM imidazole, pH 8. The 20 mM imidazole FITZAP-enriched fraction was further purified by size-exclusion chromatography (G-75 superfine; Pharmacia, Piscataway, NJ) followed by strong anion exchange chromatography (Mono-Q; Pharmacia) on an Agilent 1100 HPLC with UV detection at 220 nm.

## Recombinant lysin expression

Recombinant lysin from red and disk abalone was expressed and purified by methods from *Wilburn et al. (2018)*. Briefly, lysin coding sequences were cloned into the pET11d expression vector (Novagen), transformed into Rosetta2 chemically competent *E. coli* (EMD-Millipore, Billerica, MA) which express additional tRNA genes for Lys and Arg that are essential for lysin expression, and clones validated by Sanger sequencing (Eurofin Genomics, Louisville, KY). To provide flexibility in isotopic labeling for NMR experiments, lysin expression was performed in cultures where (1) biomass with high ribosome densities was produced by initially culturing in complex media, (2) the cells were concentrated ~4X in minimal media without nitrogen or carbon to deplete amino acid stores, (3) ammonium sulfate ($^{14}$N or $^{15}$N) and glucose (uniformly $^{12}$C or $^{13}$C) were added to regenerate amino acids with the appropriate isotopes, then (4) expression was induced by addition of IPTG. Growth under minimal media conditions provides complete removal of the N-terminal methionine from endogenous *E. coli* methionine aminopeptidase activity, leaving a single exogenous Gly on the N-terminus that has no detectable impact on lysin structure or function. Properly folded recombinant lysin was expressed into inclusion bodies that were isolated by centrifugation of cell lysate, washed to remove contaminant proteins, denatured in 5M guanidinium hydrochloride, refolded by rapid dilution, and purified using cation exchange chromatography.

## Ion exchange purification of lysin and sp18

Methods for lysin purification were adapted from *Lewis et al. (1982)*. For both natural and recombinant lysin, step chromatography was performed using CM52 cellulose (Whatman) equilibrated in 250 mM NaCl/10 mM MES/2 mM EDTA, pH 6. For recombinant lysin, methods are described above for refolding from inclusion bodies. For natural lysin, *H. rufescens* sperm were isolated by dissection of male testes and lysed by trituration in 0.1% Triton X-100/250 mM NaCl/10 mM MES/2 mM EDTA, pH 6; insoluble material (including chromatin) was removed by centrifugation at 3200 *x g* for 30 min. Crude fractions of natural or recombinant lysin were applied to a CM52 cellulose column, rinsed with >6 CVs of 250 mM NaCl/10 mM MES/2 mM EDTA, pH 6, and eluted with 3 CVs of 1 M NaCl/10 mM MES/2 mM EDTA, pH 6. Purified natural lysin and sp18 from *H. fulgens* was generously supplied by Vic Vacquier. Purified lysin and sp18 was concentrated and buffer exchanged to 150 mM NaCl/10 mM Tris, pH 7.4 using YM10 centrifugal ultrafilters (Millipore).

## Comparison of conspecific/heterospecific FITZAP-Lysin/Sp18 interactions by fluorescence polarization

Purified recombinant FITZAP proteins from red, disk, and green abalone were buffer exchanged into 0.5 mL phosphate buffered saline (PBS) using a 3 kDa centrifugal ultrafilter (Millipore), and fluorescently labeled by addition of 18 μL Alexa Fluor 488 SDP (Invitrogen, Carlsbad, CA) at 10 mg/mL DMSO at 4°C overnight with mixing. Fluorescently labeled FITZAP was separated from free fluorophore by size exclusion using Nap5 columns (GE Life Sciences, Piscataway, NJ). Protein concentrations for labeled FITZAP and unlabeled lysin were determined by BCA Protein Assay (Pierce). Each fluorescently labeled FITZAP isoform was standardized to 1 μM, lysin added to concentrations of 0, 1, 2, 5, 10, 15, 20, 30, 40, and 50 μM, and fluorescence anisotropy measured using a Fluorolog spectrofluorometer (Horiba Scientific, North Edison, NJ). All species/isoform combinations between FITZAP and lysin were performed in 150 mM NaCl/10 mM Tris, pH 7.4, and when possible, anisotropy measurements were collected in technical duplicate (although sample degradation over the course of the experiment prevented this for all combinations). For conspecific pairings with low μM binding affinities, anisotropy measurements were repeated in 500 mM NaCl/10 mM Tris, pH 7.4. Additionally, anisotropy experiments were repeated for green FITZAP isoforms with green sp18 using the same series of concentrations as lysin in 150 mM NaCl/20 mM Tris, pH 7.4. Dissociation constants ($K_d$) were estimated for all combinations by nonlinear regression using the equation

$$Anisotropy \sim \Delta A_{max} * \frac{[Lysin]+[FITZAP]+K_d - \sqrt{([Lysin]+[FITZAP]+K_d)^2 - 4*[Lysin]*[FITZAP]}}{2} + A_{intercept}$$ with the R function nlsLM in the package minpack.lm.

## Anion exchange analysis of positively charged acrosomal proteins

Preliminary experiments separating *H. rufescens* sperm lysate by anion exchange chromatography yielded the surprising result of both lysin and sp18 (highly positively charged proteins) adhering to the column and eluting at relatively high ionic strengths, and it was hypothesized that this unexpected observation may result from FITZAP supplying negative charges to these proteins as part of FITZ complexes at low ionic strength. Sperm from *H. rufescens* was isolated by dissection of testes, lysed by trituration in 2 mL of 0.1% Triton X-100/20 mM Tris, pH 8 supplemented with TURBO DNase (Ambion), and centrifuged at 2 k *x g* for 10 min. Clarified lysate was applied to a 5 mL Q Sepharose (Sigma-Aldrich) column, and 2 CV fractions collected at 0, 25, 50, 100, 200, 300, 400, and 600 mM NaCl in 20 mM Tris, pH 8. The same step gradient was performed after applying 2 mL aliquots of (1) red lysin at 0.5 mg/mL, (2) red lysin at 0.5 mg/mL with equimolar red FITZAP-8D, and (3) red lysin at 0.5 mg/mL with equimolar red FITZAP-4D. Anionic exchange fractions were separated by 15% Tris-Tricine SDS-PAGE (*Schägger and von Jagow, 1987*) and stained with Coomassie Brilliant Blue R-250.

## NMR analysis of FITZAP-Lysin interactions

Purified, isotopically labeled *H. rufescens* FITZAP-8D ($^{15}$N/$^{13}$C) was concentrated to ~0.2–1.0 mM in 50 mM NaCl/10 mM Tris, pH 7.4/7% D$_2$O using a 3 kDa centrifugal ultrafilter (Millipore). For FITZAP, all NMR experiments were performed on a Bruker Avance 800-Mhz spectrometer fitted with a TCI CryoProbe (Bruker), while NMR experiments with labeled lysin were performed on a Bruker Avance 500-Mhz spectrometer (Bruker). NMR assignments of red FITZAP-8D were obtained using a combination of 2D/3D experiments: $^{15}$N- and $^{13}$C-filtered HSQC, HNCACB, CBCAcoNH, HNCO, HNHA, $^{15}$N-HSQC-TOCSY, and $^{15}$N-HSQC-NOESY. Spectra were processed using NMRpipe (*Delaglio et al., 1995*) and analyzed using NMRFAM-SPARKY (*Lee et al., 2015*). Assignments were 70% complete for backbone atoms (91% excluding the poly-aspartate region). Chemical shift indices were calculated using TALOS-N (*Shen and Bax, 2013*). To characterize lysin binding residues, $^{15}$N- and $^{13}$C-HSQC spectra of $^{15}$N/$^{13}$C-FITZAP-8D (200 µM in 500 mM NaCl/10 mM Tris, pH 7.4/7% D$_2$O) were acquired at six concentrations of recombinant monomeric lysin (*Wilburn et al., 2018*) from 0 to 500 µM. Chemical shift perturbations (CSPs) between $^{15}$N-HSQC spectra were calculated as $\sqrt{(\Delta^1 H)^2 + (0.1 * \Delta^{15} N)^2}$. The interaction of lysin and salt with FITZAP was assessed by acquiring $^{15}$N- and $^{13}$C-HSQC spectra of $^{15}$N/$^{13}$C-FITZAP-8D (200 µM in 10 mM Tris, pH 7.4/7% D$_2$O) with or without monomeric lysin (140 µM) at different salt concentrations (150–500 mM in 50 mM steps). Reciprocal titration experiments were performed with $^{15}$N- and $^{13}$C-HSQC spectra acquired for $^{15}$N/$^{13}$C-monomeric lysin (150–200 µM in 10 mM Tris, pH 7.4/7% D$_2$O) at different salt concentrations (150 or 500 mM) with varying concentrations of FITZAP-8D (0 to 600 µM). To obtain intermolecular PRE constraints from the FITZAP-8D N-terminal ATCUN motif, R2 relaxation rates were measured for $^{15}$N- monomeric lysin (150 µM in 500 mM NaCl/10 mM Tris, pH 7.4/7% D$_2$O) with equimolar FITZAP-8D with or without 135 µM CuSO$_4$ using delays of 8.48, 16.96, 25.44, 33.92, 42.40, 50.88, and 59.36 ms. Data has been deposited in the BMRB (27962).

## Structural analysis

Using Xplor-NIH 2.48 (*Schwieters et al., 2006*; *Schwieters et al., 2003*), a structural ensemble of lysin and FITZAP-8D heterodimers was modeled by simulated annealing from 4000 to 25 K with torsion dynamics followed by Cartesian minimization using constraints from PRE measurements and degenerate CSP pairings (adapted from *Clore and Schwieters, 2003*). For comparison, a similar ensemble was constructed between lysin and VERL repeat 1 using a lysin solution structure (PDB 5utg) and a VERL repeat 1 crystal structure (PDB 5ii4) using constraints based on a cocrystal structure of lysin and VERL repeat 3 (PDB 5mr3). Figures of 3D protein models were produced using PyMOL (v.1.8, Schrodinger, LLC), regular secondary structure defined using the DSS function, and electrostatic surfaces calculated using the APBS/PDB2PQR server (*Dolinsky et al., 2004*; *Jurrus et al., 2018*).

## Characterization of FITZ complexes by dynamic light scattering

Dynamic light scattering measurements were performed using a Zetasizer Nano (Malvern Pananalytical). Natural lysin purified from red abalone testis lysate was standardized to 100 µM in 10 mM Tris, pH 7.4, and light scattering was measured with and without equimolar FITZAP-8D at different salt concentrations (150 and 500 mM NaCl). Six technical replicates of 15 scans each were collected and averaged. FITZAP-8D in isolation produced no substantial light scattering over buffer.

## Acknowledgements

The authors thank Jan Aagaard, Emily Killingbeck, Jolie Carlisle, Alberto Rivera, Richard Feldhoff, Auberon Lopez, and Harmit Malik for feedback on the research and/or manuscript. We would also like to thank Vic Vacquier for providing purified acrosomal proteins from green abalone. This work was supported by National Institutes of Health grants R01-HD076862 to WJS and K99-HD090201 to DBW.

## Additional information

### Funding

| Funder | Grant reference number | Author |
|---|---|---|
| Eunice Kennedy Shriver National Institute of Child Health and Human Development | R01-HD076862 | Willie J Swanson |
| Eunice Kennedy Shriver National Institute of Child Health and Human Development | K99-HD090201 | Damien Beau Wilburn |

The funders had no role in study design, data collection and interpretation, or the decision to submit the work for publication.

### Author contributions

Damien Beau Wilburn, Conceptualization, Data curation, Formal analysis, Funding acquisition, Validation, Investigation, Visualization, Writing - original draft, Writing - review and editing; Lisa M Tuttle, Formal analysis, Validation, Investigation, Methodology, Writing - review and editing; Rachel E Klevit, Resources, Supervision, Project administration, Writing - review and editing; Willie J Swanson, Conceptualization, Supervision, Funding acquisition, Project administration, Writing - review and editing

### Author ORCIDs

Damien Beau Wilburn (iD) https://orcid.org/0000-0002-1255-9982
Lisa M Tuttle (iD) https://orcid.org/0000-0001-8889-232X
Rachel E Klevit (iD) http://orcid.org/0000-0002-3476-969X

### Decision letter and Author response

Decision letter https://doi.org/10.7554/eLife.52628.sa1
Author response https://doi.org/10.7554/eLife.52628.sa2

## Additional files

### Supplementary files

• Supplementary file 1. FASTA sequences of FITZAP expression constructs. DNA sequences amplified by fusion PCR and cloned into pET11d for recombinant bacterial expression.

• Transparent reporting form

## Data availability

Sequences have been deposited into Genbank under Accession # MN102340-MN102343 and NMR data have been deposited in the BMRB under accession code 27962.

The following datasets were generated:

| Author(s) | Year | Dataset title | Dataset URL | Database and Identifier |
|---|---|---|---|---|
| Wilburn DB, Tuttle LM, Klevit RE, Swanson WJ | 2019 | NMR assignments for H rufescens FITZAP 8D | http://www.bmrb.wisc.edu/data_library/summary/index.php?bmrbId=27962 | Biological Magnetic Resonance Data Bank, 27962 |
| Wilburn DB, Tuttle LM, Klevit RE, Swanson WJ | 2019 | Haliotis walallensis sperm protein 18kDa (sp18-1) mRNA, complete cds | https://www.ncbi.nlm.nih.gov/nuccore/MN102340 | NCBI Genbank, MN102340 |
| Wilburn DB, Tuttle LM, Klevit RE, Swanson WJ | 2019 | Haliotis discus sperm protein 18kDa (sp18-2) mRNA, complete cds | https://www.ncbi.nlm.nih.gov/nuccore/MN102341 | NCBI Genbank, MN102341 |
| Wilburn DB, Tuttle LM, Klevit RE, Swanson WJ | 2019 | Haliotis cracherodii sperm protein 18kDa (sp18-3) mRNA, complete cds | https://www.ncbi.nlm.nih.gov/nuccore/MN102342 | NCBI Genbank, MN102342 |
| Wilburn DB, Tuttle LM, Klevit RE, Swanson WJ | 2019 | Haliotis walallensis FITZ anionic partner 6D precursor (FITZAP-6D) mRNA, complete cds | https://www.ncbi.nlm.nih.gov/nuccore/MN102343 | NCBI Genbank, MN102343 |

The following previously published datasets were used:

| Author(s) | Year | Dataset title | Dataset URL | Database and Identifier |
|---|---|---|---|---|
| Vacquier VD, Carner KR, Stout CD | 1990 | H.rufescens sperm lysin mRNA, complete cds | https://www.ncbi.nlm.nih.gov/nuccore/M34388 | NCBI Genbank, M34388 |
| Vacquier VD, Carner KR, Stout CD | 1990 | Haliotis walallensis stearns sperm lysin mRNA, complete cds | https://www.ncbi.nlm.nih.gov/nuccore/M59969 | NCBI Genbank, M59969 |
| Lee Y-H, Vacquier VD | 1992 | Haliotis kamtschatkana kamtschatkana lysin mRNA, complete cds | https://www.ncbi.nlm.nih.gov/nuccore/M59970 | NCBI Genbank, M59970 |
| Lee Y-H, Vacquier VD | 1992 | Haliotis cracherodi lysin mRNA, complete cds | https://www.ncbi.nlm.nih.gov/nuccore/M59971 | NCBI Genbank, M59971 |
| Lee Y-H, Vacquier VD | 1992 | Haliotis fulgens lysin mRNA, complete cds | https://www.ncbi.nlm.nih.gov/nuccore/M59972 | NCBI Genbank, M59972 |
| Lee YH, OtaT, Vacquier VD | 1995 | Haliotis discus hannai sperm lysin mRNA, complete cds | https://www.ncbi.nlm.nih.gov/nuccore/M98875 | NCBI Genbank, M98875 |
| Galindo BE, Moy GW, Swanson WJ, Vacquier VD | 2002 | Haliotis rufescens vitelline envelope sperm lysin receptor (VERL) mRNA, partial cds | https://www.ncbi.nlm.nih.gov/nuccore/AF453553 | NCBI Genbank, AF453553 |
| Galindo BE, Vacquier VD, Swanson WJ | 2003 | Haliotis kamtschatkana vitelline envelope sperm lysin receptor gene, partial cds | https://www.ncbi.nlm.nih.gov/nuccore/AF490761 | NCBI Genbank, AF490761 |
| Galindo BE, Vacquier VD, Swanson WJ | 2003 | Haliotis walallensis vitelline envelope sperm lysin receptor gene, partial cds | https://www.ncbi.nlm.nih.gov/nuccore/AF490762 | NCBI Genbank, AF490762 |
| Galindo BE, Vacquier VD, Swanson WJ | 2003 | Haliotis discus hannai vitelline envelope sperm lysin receptor gene, partial cds | https://www.ncbi.nlm.nih.gov/nuccore/AF490763 | NCBI Genbank, AF490763 |
| Galindo BE, Vacquier VD, Swanson WJ | 2003 | Haliotis cracherodii vitelline envelope sperm lysin receptor gene, partial cds | https://www.ncbi.nlm.nih.gov/nuccore/AF490765 | NCBI Genbank, AF490765 |
| Galindo BE, Vacquier VD, Swanson WJ | 2003 | Haliotis fulgens vitelline envelope sperm lysin receptor gene, partial cds | https://www.ncbi.nlm.nih.gov/nuccore/AF490766 | NCBI Genbank, AF490766 |
| Swanson WJ, Vacquier VD | 1995 | Haliotis rufescens fertilization protein mRNA, complete cds | https://www.ncbi.nlm.nih.gov/nuccore/L36552 | NCBI Genbank, L36552 |

| | | | | |
|---|---|---|---|---|
| Swanson WJ, Vacquier VD | 1995 | Haliotis assimilis fertilization protein mRNA, complete cds | https://www.ncbi.nlm.nih.gov/nuccore/L36554 | NCBI Genbank, L36554 |
| Swanson WJ, Vacquier VD | 1995 | Haliotis fulgens fertilization protein mRNA, complete cds | https://www.ncbi.nlm.nih.gov/nuccore/L36589 | NCBI Genbank, L36589 |
| Palmer MR, McDowall MH, Stewart L, Ouaddi A, Maccoss MJ, Swanson W | 2013 | Haliotis rufescens isolate 8D sperm protein 6kDa mRNA, partial cds | https://www.ncbi.nlm.nih.gov/nuccore/KC752594 | NCBI Genbank, KC752594 |
| Palmer MR, McDowall MH, Stewart L, Ouaddi A, Maccoss MJ, Swanson WJ | 2013 | Haliotis discus isolate 11D sperm protein 6kDa mRNA, complete cds | https://www.ncbi.nlm.nih.gov/nuccore/KC752595 | NCBI Genbank, KC752595 |
| Palmer MR, McDowall MH, Stewart L, Ouaddi A, Maccoss MJ, Swanson WJ | 2013 | Haliotis kamtschatkana isolate 4Ds sperm protein 6kDa mRNA, complete cds | https://www.ncbi.nlm.nih.gov/nuccore/KC752597 | NCBI Genbank, KC752597 |
| Palmer MR, McDowall MH, Stewart L, Ouaddi A, Maccoss MJ, Swanson WJ | 2013 | Haliotis rufescens isolate 4D sperm protein 6kDa mRNA, complete cds | https://www.ncbi.nlm.nih.gov/nuccore/KC752598 | NCBI Genbank, KC752598 |
| Palmer MR, McDowall MH, Stewart L, Ouaddi A, Maccoss MJ, Swanson WJ | 2013 | Haliotis fulgens isolate 4D sperm protein 6kDa mRNA, complete cds | https://www.ncbi.nlm.nih.gov/nuccore/KC752599 | NCBI Genbank, KC752599 |
| Palmer MR, McDowall MH, Stewart L, Ouaddi A, Maccoss MJ, Swanson WJ | 2013 | Haliotis fulgens isolate 1D sperm protein 6kDa mRNA, complete cds | https://www.ncbi.nlm.nih.gov/nuccore/KC752600 | NCBI Genbank, KC752600 |
| Palmer MR, McDowall MH, Stewart L, Ouaddi A, Maccoss MJ, Swanson WJ | 2013 | Haliotis kamtschatkana isolate 4Dl sperm protein 6kDa mRNA, complete cds | https://www.ncbi.nlm.nih.gov/nuccore/KC752601 | NCBI Genbank, KC752601 |
| Palmer MR, McDowall MH, Stewart L, Ouaddi A, Maccoss MJ, Swanson WJ | 2013 | Haliotis cracherodii sperm protein 6kDa mRNA, complete cds | https://www.ncbi.nlm.nih.gov/nuccore/KC752602 | NCBI Genbank, KC752602 |
| Wilburn DB, Tuttle LM, Klevit RE, Swanson WJ | 2018 | Red abalone lysin F104A | http://www.bmrb.wisc.edu/data_library/summary/index.php?bmrbId=30246 | Biological Magnetic Resonance Data Bank, 30246 |

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
