## [Decision Letter]

**Acceptance summary:**

This paper reports an unusual function of an intrinsically disordered domain in facilitating storage of fertilization proteins in abalone sperm and in their release upon exposure to sea water. The authors combine NMR spectroscopy with fluorescence polarization and dynamic light scattering to show that the highly charged and intrinsically disordered domain organizes the storage of a fertilization protein at extremely high concentration in sperm, which is required for egg penetration. The malleable and disordered domain condenses fertilization proteins in soluble form by targeting the same interaction interface as the egg protein and by compensating repulsive electrostatic forces in oligomeric assemblies through provision of opposite charges. Upon release into sea water, high concentrations of salt screen charge interactions thus facilitating disassembly and fertilization. Besides uncovering an intriguing mechanism of soluble protein condensation, the authors show that rapid co-evolution can proceed indirectly through protein-protein interaction networks. The study is of broad interest because it integrates topics of intrinsic protein disorder, evolutionary biology, protein biochemistry and biophysics.

**Decision letter after peer review:**

Thank you for submitting your article "Indirect sexual selection drives rapid sperm protein evolution in abalone" for consideration by *eLife*. Your article has been reviewed by two peer reviewers, including Hannes Neuweiler as the Reviewing Editor and Reviewer #1, and the evaluation has been overseen by Patricia Wittkopp as the Senior Editor. The following individual involved in review of your submission has also agreed to reveal their identity: Kristaps Jaudzems (Reviewer #2).

The reviewers have discussed the reviews with one another and the Reviewing Editor has drafted this decision to help you prepare a revised submission.

Review synthesis:

The manuscript "Indirect sexual selection drives rapid sperm protein evolution in abalone" by Wilburn et al. reports on the discovery and characterization of the functional role of the small, intrinsically disordered domain sp6 in storage and release of the abalone sperm protein lysin, which participates in fertilization. Wilburn et al. use NMR spectroscopy, fluorescence polarization and dynamic light scattering to investigate structure and interaction of sp6 bound to lysin in-vitro and its response to changes in salt concentration. The dependence of complex formation on salt concentration explains liberation of lysin upon exocytosis into seawater. The authors propose a mechanism whereby the negatively charged sp6 binds to the positively charged lysin and generates a zwitter-ionic complex that allows storage of lysin at extraordinary high concentrations. Solubility is facilitated by formation of large, multimeric assemblies that are held together by attractive charge-charge interactions. Given the novel functional role the authors rename sp6 as fuzzy interacting transient zwitterion anionic partner (FITZAP). Using NMR spectroscopy they find that the binding interface of lysin is common for both, FITZAP and the egg vitelline envelope receptor of lysin (VERL). Determination of binding affinities between FITZAP and lysin from three different species probed by fluorescence polarization indicates similar species-specificity as lysin-VERL interactions. Based on their findings the authors propose an indirect sexual selection mechanism resulting in co-evolution of FITZAP to maintain binding of lysin, which co-evolves with VERL via direct sexual selection. The mechanism is proposed to generally explain rapid sperm protein evolution.

Reproductive proteins are encoded by fast evolving genes. Yet, rapid coevolution of fertilization proteins that are not directly involved in interactions with the egg is puzzling. Moreover, it is a mystery how sperm can store lysin at internal concentrations approaching 1 M required for successful fertilization. The study by Wilburn et al. provides new interesting details about the binding mode and interaction specificities of the FITZAP proteins allowing the authors to elucidate function and solubility within the sperm acrosome, and to explain their rapid evolution even though they are not directly involved in fertilization. The study is carefully designed and experiments are thoroughly performed. Furthermore, the manuscript is well written and results are properly interpreted.

However, the following issues need to be addressed before publication:

1) The authors claim to demonstrate how the extraordinary packaging of lysin and sp18 within the acrosome is achieved. However, while the salt conditions used for NMR studies correspond to intracellular levels, the protein concentrations were much lower than intracellular. Therefore, the obtained results may not fully depict the protein complexes that are formed within the acrosome. Is it possible to concentrate the complex to at least 10 mM concentration and record NMR data to see, which residues lose intensity?

2) DLS measurements indicate formation of oligomers with mean diameter of 400 nm, which does not fit with the heterodimeric model presented in Figure 1C. Can the NMR data be interpreted in a way to describe the oligomer structure that should have more than one interaction interface?

3) It would be interesting to see what the 400 nm diameter oligomers look like under an electron microscope, especially because the size distribution is rather homogeneous.

4) At the beginning of the Results section the authors show that FITZAP has a molecular weight of 3-4 kDa (35 residues, Figure 2—figure supplement 1) and argue for it being an intrinsically disordered domain. But given the small size of FITZAP one may argue that it is actually a peptide rather than a domain. Most peptides are unfolded. Is it really appropriate to assign FITZAP to the class of intrinsically disordered domains?

5) In the Results section, paragraph two, the authors argue that incubation of lysin with FITZAP would lead to binding of the complex to an anion exchange column and retard lysin. But in Figure 1—figure supplement 3 there is hardly any retardation evident. Samples from higher ionic strength eluates show hardly protein bands in SDS-PAGE (hardly visible in the gel images) and, if at all, traces of a complex. This appears reasonable because the total number of negative charges of FITZAP does not fully compensate the higher number of positive charges of lysin.

6) Figure 2—figure supplement 1: The authors argue that FITZAP is disordered when free and bound to lysin. But what are the concentrations probed in NMR data shown in this figure? This should be stated in the figure caption. Are the authors sure that they saturated binding of FITZAP to lysin by applying sufficient excess concentration of lysin?

7) Figure 1C, lower panel: in the structural model the dimensions of FITZAP-8D and lysin appear similar. This is confusing because FITZAP is a 35-residue peptide (or domain) while lysin is a 125-residue domain.

8) In paragraph three of the Results section (Figure 3—figure supplement 1) the authors infer kinetics of FITZAP-lysin association/dissociation from NMR chemical shift perturbation. But this is not valid. No accurate kinetic data can be inferred from the data shown. In order to measure kinetics the authors would need to carry out rapid-mixing experiments (possibly requiring a stopped-flow machine) using e.g. fluorescence polarization as a probe for association/dissociation. Rate constants of dissociation could be measured using chasing experiments in combination with rapid-mixing. Anyway, in my opinion, no support from kinetic data is required for the conclusions of this work. It is reasonable to assume that the rate constant of dissociation of the complex, once it is released from sperm to sea water environment, is substantially faster than diffusion and binding of lysin to the egg cell receptor VERL.

9) In the Results paragraph ten the authors state that lysin has a higher affinity for high-D FITZAP isoforms compared to low-D isoforms. This finding suggests that not only hydrophobics but also the charged tail of FITZAP plays a role in binding to lysin. Lysin and FITZAP are net oppositely charged and will attract each other in solution, which will lead to electrostatically assisted binding (see e.g. Schreiber and Fersht, Nat Str Biol 1996, 3, 427-431; Janin, Proteins 1997, 28, 153-161; Schreiber et al., Chem Rev 2009, 109, 839-860).

10) In paragraph one of the Discussion the authors state that they anticipate minimal evolutionary selection on the primary sequence of FITZAP because it is a disordered domain with low sequence variation. Does this not contradict their main conclusion saying that indirect sexual selection acts on FITZAP?

11) A general point: the density of data shown in main figures is rather low. The manuscript was probably originally written up concisely for a journal with very limited space. The authors could move supplemental figures into the main manuscript, which would be beneficial for the reader and improve the overall quality.

---

## [Author Response]

However, the following issues need to be addressed before publication:1) The authors claim to demonstrate how the extraordinary packaging of lysin and sp18 within the acrosome is achieved. However, while the salt conditions used for NMR studies correspond to intracellular levels, the protein concentrations were much lower than intracellular. Therefore, the obtained results may not fully depict the protein complexes that are formed within the acrosome. Is it possible to concentrate the complex to at least 10 mM concentration and record NMR data to see, which residues lose intensity?

We have been unable to concentrate lysin beyond ~1mM. Experiments conducted using low molecular weight cutoff centrifugal ultrafilters yielded the same final concentration of lysin regardless of salt concentration and/or presence of equimolar FITZAP. This may be due to fuzzy condensates being disrupted upon centrifugation and/or sufficient exchange of FITZ complexes still permitting lysin aggregation. However, these concentrations were not required to detect both chemical shift and intensity changes by ^15^N-HSQC (more intensity loss in low salt, more CSPs in high salt, due to differences in exchange rate, see Figure 3—figure supplement 1).

2) DLS measurements indicate formation of oligomers with mean diameter of 400 nm, which does not fit with the heterodimeric model presented in Figure 1C. Can the NMR data be interpreted in a way to describe the oligomer structure that should have more than one interaction interface?

The model displayed in Figure 1C (now Figure 2C) reflects the dimeric state of lysin-FITZAP based on CSPs measured in high salt. The available NMR data can partially describe the larger oligomeric condition. In a titration series of labeled FITZAP with unlabeled lysin at different salt concentrations, we only observe intensity loss in the poly-aspartate region under low salt conditions, while there is consistent intensity loss (and CSPs) in the more hydrophobic C-terminal region of FITZAP regardless of salt concentration. Combined with the DLS data, this supports that hydrophobic interactions facilitate the initial heterodimer formation while ionic interactions enable the formation of the larger oligomeric species. These data are summarized in Figure 3. Reciprocal experiments with labeled lysin and unlabeled FITZAP at different salt conditions primarily report on the initial heterodimeric condition. While we see the same lysin residues being perturbed in both low and high salt, the formation of high molecular weight FITZ complexes results in line broadening beyond detection, which is supported by the ~10% median intensity loss in 150 vs 500 mM NaCl (Figure 3—figure supplement 1A).

3) It would be interesting to see what the 400 nm diameter oligomers look like under an electron microscope, especially because the size distribution is rather homogeneous.

We agree this would be interesting, but is likely not possible due to the dynamic nature of FITZ complexes. As our NMR experiments support that the FITZ complexes are transient and rapidly exchanging, it would be difficult to find equilibrium conditions where they would stably adhere to an EM grid. Furthermore, the fuzzy nature of lysin-FITZAP interactions likely would preclude any form of particle selection and averaging to gain additional structural insights.

4) At the beginning of the Results section the authors show that FITZAP has a molecular weight of 3-4 kDa (35 residues, Figure 2—figure supplement 1) and argue for it being an intrinsically disordered domain. But given the small size of FITZAP one may argue that it is actually a peptide rather than a domain. Most peptides are unfolded. Is it really appropriate to assign FITZAP to the class of intrinsically disordered domains?

Given its length, a case could be made for designating FITZAP as a peptide instead of a protein, yet in either scenario it is still an IDP (intrinsically disordered protein/peptide). As domains are defined based on a folded 3D structure, IDPs by definition lack domains and we do not use that term in the manuscript. While FITZAP likely falls in the grey area between proteins and peptides, we favor the use of “protein” as it is a naturally synthesized polypeptide chain with a now defined function.

5) In the Results section, paragraph two, the authors argue that incubation of lysin with FITZAP would lead to binding of the complex to an anion exchange column and retard lysin. But in Figure 1—figure supplement 3 there is hardly any retardation evident. Samples from higher ionic strength eluates show hardly protein bands in SDS-PAGE (hardly visible in the gel images) and, if at all, traces of a complex. This appears reasonable because the total number of negative charges of FITZAP does not fully compensate the higher number of positive charges of lysin.

This is an excellent point and further explanation has been added to the Results section. The difference between the ex vivo and in vitro results likely has to do with sample preparation, initial protein concentration, and the FITZ complex dissociation rate. Through several experiments to optimize the results, we found that both retention of lysin and sp18 on the anion exchange column was partially dependent on (1) length of centrifugation prior to applying the sample to the column (where FITZ complexes were presumably large enough to sediment), and (2) time to complete the chromatography (where less lysin/sp18 would be retained the longer they were on-column). To maximize the observed effect, sperm lysates were prepared by trituration of sperm in a small volume of Tris-buffered detergent solutions lacking NaCl to minimize ionic disruption of complexes, addition of DNaseI to reduce the sample viscosity, and a very brief centrifugation step to remove only the largest insoluble particles. By lysing in a near-minimal volume in the absence of salt, FITZ complexes should both be near their maximum concentration and experience slower dissociation. In contrast, our in vitro reconstitutions are limited by the solubility of lysin, such that we never achieve the extreme conditions of the naturally purified samples. The detection of any retardation under conditions orders of magnitude below the physiological concentrations supports that our ex vivo results are biologically meaningful.

6) Figure 2—figure supplement 1: The authors argue that FITZAP is disordered when free and bound to lysin. But what are the concentrations probed in NMR data shown in this figure? This should be stated in the figure caption. Are the authors sure that they saturated binding of FITZAP to lysin by applying sufficient excess concentration of lysin?

Figure 2—figure supplement 1 is not at saturation conditions, as it is not possible to saturate FITZAP and collect triple resonance spectra due to both lysin solubility limits and line broadening of FITZAP backbone amides at the binding interface. However, we have now added an additional supplemental figure (Figure 2—figure supplement 2) of FITZAP 13C-HSQC with different concentrations of lysin, including one near saturation (92%). Notably few 13C chemical shifts are observed in the CA-HA region, with the largest being A27, M34, and F35 before they broaden/overlap beyond detection. In both the CB-HB and methyl regions, CSPs are only observed in the 1H dimension, corroborating that FITZAP does not adopt secondary structure in the presence of lysin. Furthermore, based on our model of the lysin-FITZAP fuzzy heterodimer, these perturbed residues are localized near several aromatic residues on lysin (including H61, W62, Y65, and W68) such that changes in CA/HA chemical shifts may be due to ring current effects and broadening from chemical exchange.

7) Figure 1C, lower panel: in the structural model the dimensions of FITZAP-8D and lysin appear similar. This is confusing because FITZAP is a 35-residue peptide (or domain) while lysin is a 125-residue domain.

This is a consequence of the more linear FITZAP being compared to the more globular lysin.

8) In paragraph three of the Results section (Figure 3—figure supplement 1) the authors infer kinetics of FITZAP-lysin association/dissociation from NMR chemical shift perturbation. But this is not valid. No accurate kinetic data can be inferred from the data shown. In order to measure kinetics the authors would need to carry out rapid-mixing experiments (possibly requiring a stopped-flow machine) using e.g. fluorescence polarization as a probe for association/dissociation. Rate constants of dissociation could be measured using chasing experiments in combination with rapid-mixing. Anyway, in my opinion, no support from kinetic data is required for the conclusions of this work. It is reasonable to assume that the rate constant of dissociation of the complex, once it is released from sperm to sea water environment, is substantially faster than diffusion and binding of lysin to the egg cell receptor VERL.

The chemical shift is itself a rate that must be interpreted relative to the chemical exchange rate between free and bound states, hence differences in observed NMR peak centers and linewidths in fast (Δω << k_ex_), intermediate (Δω ~ k_ex_), and slow (Δω >> k_ex_) exchange regimes. Comparing changes in observed chemical shift and peak intensities/linewidths is informative of the relative exchange regime (see Dwek, 1973). Changes in observed chemical shift are more pronounced under high salt compared to greater intensity loss under low salt, suggesting a change in k_ex_ (assuming similar Δω’s under both conditions, since presumably the lysin backbone amides are experiencing the same altered chemical environment upon hydrophobic packing). We make no specific references to the absolute magnitude of this change, only their relative qualitative difference.

9) In the Results paragraph ten the authors state that lysin has a higher affinity for high-D FITZAP isoforms compared to low-D isoforms. This finding suggests that not only hydrophobics but also the charged tail of FITZAP plays a role in binding to lysin. Lysin and FITZAP are net oppositely charged and will attract each other in solution, which will lead to electrostatically assisted binding (see e.g. Schreiber and Fersht, Nat Str Biol 1996, 3, 427-431; Janin, Proteins 1997, 28, 153-161; Schreiber et al., Chem Rev 2009, 109, 839-860).

We agree that both hydrophobic and ionic interactions facilitate lysin-FITZAP interactions to associate into high molecular weight FITZ complexes, and this is discussed in paragraph three of the Results section. The text has also been expanded in paragraph four. The length of the poly-D array alone does not predict lysin-FITZAP association by FP (e.g. Red Lysin binds Red FITZAP-8D efficiently, but not Disk FITZAP-11D; similarly, Green Lysin binds Green FITZAP-4D but not Red FITZAP-4D). This is consistent with a two-step binding event where sequence-specific hydrophobic interactions result in lysin-FITZAP heterodimers that then electrostatically associate into higher order FITZ complexes.

10) In paragraph one of the Discussion the authors state that they anticipate minimal evolutionary selection on the primary sequence of FITZAP because it is a disordered domain with low sequence variation. Does this not contradict their main conclusion saying that indirect sexual selection acts on FITZAP?

We have better clarified this point in the text. As an IDP, FITZAP likely lacks strong intramolecular epistatic purifying selection which most protein-coding genes experience to maintain a tertiary fold (and hence its function); instead, most of its evolution is likely due to intermolecular coevolution with binding partners.

11) A general point: the density of data shown in main figures is rather low. The manuscript was probably originally written up concisely for a journal with very limited space. The authors could move supplemental figures into the main manuscript, which would be beneficial for the reader and improve the overall quality.

Supplemental figures 3A and 5 have now been moved to the main text (now Figure 1 and 3).